# GLP-1 Receptor Agonists in Neurodegeneration: Neurovascular Unit in the Spotlight

**DOI:** 10.3390/cells11132023

**Published:** 2022-06-25

**Authors:** Giulia Monti, Diana Gomes Moreira, Mette Richner, Henricus Antonius Maria Mutsaers, Nelson Ferreira, Asad Jan

**Affiliations:** 1Department of Biomedicine, Aarhus University, Høegh-Guldbergs Gade 10, DK-8000 Aarhus, Denmark; gm@biomed.au.dk (G.M.); mette.richner@biomed.au.dk (M.R.); ncgferreira@gmail.com (N.F.); 2VIB-KU Leuven Center for Brain & Disease Research, Herestraat 49, 3000 Leuven, Belgium; dianaraquel.gomesmoreira@kuleuven.be; 3Department of Clinical Medicine, Aarhus University, Palle Juul-Jensens Boulevard 82, DK-8200 Aarhus, Denmark; h.a.m.mutsaers@clin.au.dk

**Keywords:** glucagon-like peptide-1 (GLP-1), Alzheimer’s disease, Parkinson’s disease, neurodegeneration, neurovascular unit

## Abstract

Defects in brain energy metabolism and proteopathic stress are implicated in age-related degenerative neuronopathies, exemplified by Alzheimer’s disease (AD) and Parkinson’s disease (PD). As the currently available drug regimens largely aim to mitigate cognitive decline and/or motor symptoms, there is a dire need for mechanism-based therapies that can be used to improve neuronal function and potentially slow down the underlying disease processes. In this context, a new class of pharmacological agents that achieve improved glycaemic control via the glucagon-like peptide 1 (GLP-1) receptor has attracted significant attention as putative neuroprotective agents. The experimental evidence supporting their potential therapeutic value, mainly derived from cellular and animal models of AD and PD, has been discussed in several research reports and review opinions recently. In this review article, we discuss the pathological relevance of derangements in the neurovascular unit and the significance of neuron–glia metabolic coupling in AD and PD. With this context, we also discuss some unresolved questions with regard to the potential benefits of GLP-1 agonists on the neurovascular unit (NVU), and provide examples of novel experimental paradigms that could be useful in improving our understanding regarding the neuroprotective mode of action associated with these agents.

## 1. Introduction

The glucagon-like peptide-1 (GLP-1) is a biological product derived from the post-translational processing of proglucagon. It is secreted by specific cell populations in the gastrointestinal tract (the enteroendocrine L cells), but also by some neuronal populations in the hindbrain (the nucleus of solitary tract-NTS: an important region involved in energy intake) [1,2,3,4]. The physiological stimulus for the endogenous GLP-1 secretion is nutrient ingestion and consequent rise in plasma glucose, which leads to a two- to three-fold enhancement in the circulating GLP-1 compared to basal levels [5,6,7]. This is rapidly followed by an incretin effect (see Glossary) that restores the physiological levels of glucose in circulation [1,3]. The nascent GLP-1 is 37 amino acids in length and has two naturally bioactive forms in circulation, GLP-1 (7–37) and GLP-1 (7–36)-amide [7,8] (Figure 1A). The circulating GLP-1 is rapidly cleared from the circulation due to the proteolytic degradation by the serine protease dipeptidyl peptidase (DPP-4) and neutral endopeptidase 24.11 (NEP 24.11; also known as neprilysin) [1,7]. The direct biological actions of native bioactive forms of GLP-1 are mediated by the activation of the GLP-1 receptor, which belongs to the family B of the G-protein-coupled receptors (GPCRs) [1,4,7,9]. Studies on mRNA and protein expression indicate that GLP-1 receptors are distributed throughout several tissues including the pancreas, kidney, stomach, heart and brain [1,3,10,11,12,13]. In the brain, GLP-1 and/or GLP-1 receptor mRNA have been detected in some regions of the thalamus, hypothalamus, hippocampal pyramidal neurons, cortex, cerebellar Purkinje cells and in the brainstem [11,14,15,16,17]. The immediate effects of GLP-1 in the cells expressing GLP-1 receptors are attributed to the rise in intracellular cAMP levels and protein kinase A (PKA) activation (Figure 1B), which, in pancreatic β-cells, is directly responsible for calcium influx and insulin secretion [1,2,7,18,19]. Despite the short half-life of the endogenous GLP-1, the interplay of peripheral and central GLP-1 receptor signalling helps achieve glycaemic control and energy intake by slowing gastric emptying and reducing caloric intake (Figure 2) [2,20]. In this context, the gut-derived GLP-1 crosses the blood–brain barrier (BBB) and binds to receptors in the circumventricular organs of the brainstem, transmitting metabolic information to the neurons responsible for feeding behaviour (e.g., in the NTS and hypothalamus) [2,10,20,21]. Furthermore, GLP-1 signalling in the brain is also implicated in cognitive functions, based on the studies showing improved learning and memory performance in rats overexpressing GLP-1 receptors in the hippocampus, or the restoration of learning deficits in GLP-1 receptor-deficient mice following hippocampal *Glp1r* gene transfer [14,22].

The discovery of the insulinotropic glucose-lowering effects of GLP-1 was a major breakthrough that formed the basis for developing the first generation of synthetic GLP-1 analogues and, subsequently, their approval for the treatment of Type 2 diabetes mellitus (T2DM) and obesity in 2005 [2,3]. Since then, a number of DPP-4 degradation-resistant and long-acting synthetic GLP-1 receptor agonists (GLP-1RAs) have been developed that differ in chemical modifications and duration of action (Table 1; see references [2,23] for further details). In brief, the short-acting drugs include lixisenatide and exenatide (Byetta), while the long-acting GLP-1RAs are exemplified by liraglutide, semaglutide, exenatide (Bydureon) and dulaglutide. The main pharmacodynamic difference between the two classes is that the short-acting agonists lower postprandial glucose by delaying gastric emptying, while the long-acting GLP-1RAs have a stronger incretin effect [1,23,24]. Since the short-acting GLP-1RAs are more rapidly cleared from plasma, GLP-1 receptor activation is also short-lived. In contrast, the long-acting GLP-1RAs reach a steady state concentration, with minor fluctuations between the doses, thus causing a continuous stimulation of GLP-1 receptors [3,24]. As a result of this difference, long-acting GLP-1RAs are more effective in lowering fasting plasma glucose and in improving glycaemic status compared to the short-acting compounds, as observed by lower levels of glycated haemoglobin-HbA1c: an index of glycaemic control; see Glossary) [2,25].

The advent of GLP-1RAs led to further discoveries that sustained GLP-1RAs exposure is associated with pro-survival cellular gene expression, purportedly mediated by cAMP responsive element binding (CREB) signalling that involves increased expression and/or activity of receptor tyrosine kinases PI3K/AKT, epidermal growth factor receptor (EGFR) and hypoxia-inducible factor 1-alpha (HIF-1α) (Figure 1B) [1,4,7,26]. Through these latter mechanisms, GLP-1RAs promote pancreatic β-cell neogenesis, stimulate cell growth and increase insulin synthesis in the β-cells [18,23,27,28]. Furthermore, a growing list of research studies show that GLP-1RAs exert cytoprotective and anti-inflammatory effects in cultured cells and in vivo. For instance, GLP-1RAs modulate autophagy [29,30,31], reduce endoplasmic reticulum stress (in response to glucotoxicity and lipotoxicity) [32,33,34], and induce changes in gene expression that lead to a favourable metabolic reprogramming and redox homeostasis [4,26,35,36,37]. Burgeoning evidence also indicates that GLP-1RAs have neuroprotective properties that can potentially be translated into novel therapies for neurological conditions, in particular neurodegeneration and ischaemic brain injury [14,17,22,38,39]. In a broader context, the long-term enhancement of GLP-1 signalling leads to functional improvements across several tissues including the liver, kidney, heart, skeletal muscle and vascular smooth muscle independent of the incretin effect [3,4,7]. 

In this article, we intend to focus on GLP-1RAs in the context of neurodegenerative diseases using examples from Alzheimer’s disease (AD) and Parkinson’s disease (PD), since there is a growing list of preclinical studies showing GLP-1RAs as potential disease-modifying agents in these disorders. In brief, we discuss the neuroprotective and anti-inflammatory effects of GLP-1RAs in the context of neurovascular function and metabolic homeostasis between neurons, astroglia and local elements within the neurovascular unit (NVU). In the sections below, first we provide an overview of the salient findings with regard to the use of GLP1-RAs in experimental models of AD and PD. Then, we introduce the concepts concerning neurovascular coupling and brain energy metabolism, followed by a discussion on relevant aspects in AD and PD. Lastly, we discuss the evidence in support of the hypothesis that the neuroprotective effects of long-term GLP-1RAs may partly be mediated by improved neurovascular coupling, and conclude with outlining some aspects as the outstanding questions.

## 2. GLP-1RAs in Neurodegenerative Diseases

Late-onset AD (also termed sporadic AD) is the most common neurodegenerative disease and the leading cause of dementia in the elderly population worldwide [40,41], whereas idiopathic PD is the most common neurodegenerative cause of motor disability in the population over 65 years of age [42]. Among the risk factors for sporadic AD and idiopathic PD, advancing age represents the most important risk factor in the vast majority (>85%) of cases, followed by a positive family history and head trauma [40,43]. Clinically, the presenting features of these diseases are remarkably distinct such that AD patients exhibit significant disturbances in recent memory, which eventually progresses to visuospatial disorientation and profound disability in performing routine everyday tasks [41]. In contrast, the presenting features in clinical PD include the slowness of movement initiation and maintenance of posture [44,45]. However, long-standing (>15–20 years) PD is also associated with cognitive decline and dementia [44,46]. The detailed overview of the putative mechanisms underlying neurodegeneration in AD and PD is beyond the scope of this article, and can be found in relevant review opinions elsewhere [42,47,48]. For the present discussion, it suffices to mention that the neuropathology of both AD and PD share certain features, which supports an aetiological role of the proteopathic stress due to protein aggregation and energy dyshomeostasis in the ageing brain. In particular, both diseases are associated with: (i) intracerebral proteinaceous deposits across several cortical and subcortical regions, i.e., amyloid Aβ plaques and neurofibrillary tangles (comprised of hyperphosphorylated tau) in AD, and Lewy-related alpha-synuclein pathology in PD; (ii) neuronal loss within specific cell populations and pathological alterations in neuropil that dictate the clinical outcome (i.e., entorhinal cortex and hippocampus in AD and nigrostriatal circuitry in PD); and (iii) pathological alterations within glial cells, including pro-inflammatory phenotypes [40,47,49]. 

GLP-1RAs have emerged as promising therapeutic candidates for tackling neuronal loss and neuroinflammation since GLP-1 signalling seems to have beneficial effects on several factors implicated in the pathogenesis of neurodegenerative diseases, e.g., ER stress, impaired redox homeostasis, autophagy and chronic inflammation (reviewed in [2,4,17,22,50,51,52]). In this context, the existing literature (predominantly based on the use of liraglutide or exendin-4) maintains that GLP-1RAs promote neurogenesis, enhance neuronal survival and synaptogenesis, attenuate neuroinflammation and/or reduce the pathological markers of protein aggregation in animal models of neuronal injury [17,53,54,55,56,57,58,59,60,61,62,63,64,65,66,67,68,69,70,71,72,73]. For instance, in cultures of embryonic cortical neurons, exendin-4 promoted neurite growth and protected against cell death induced by hypoxia and the mitochondrial toxin 6-hydroxydopamine (6-OHDA) [63]. Also in neuronal cultures, liraglutide treatment mitigated the synaptotoxicity of Aβ oligomers and reduced the tau hyperphosphorylation induced by Aβ1-42 [67,74]. Similarly, liraglutide treatment in cultures of neuroblastoma cells exposed to methylglyoxal (a potent glycation stress inducer) enhanced cell viability and reduced cytotoxicity [75]. Besides in vitro evidence, GLP-1RAs have been reported to exert neuroprotective effects in diverse animal models of neuronal damage resulting from acute CNS injury (stroke, traumatic brain injury), sub-acute loss of dopaminergic neurons (chemical parkinsonism; based on injections of mitochondrial toxins 6-OHDA or 1-methyl-4-phenyl-1,2,3,6-tetrahydropyridine, MPTP) and chronic proteopathic stress (transgenic models of Aβ amyloidosis and tauopathy) [22,51] (Table 2). In transgenic rodent models of AD, liraglutide or exendin-4 treatment has been associated with a reduction in the amyloid plaque burden [62,72,76,77,78] and tau phosphorylation [56,58,67,72,73,79], which are considered invariable pathogenic factors underlying neurodegeneration in AD. Moreover, liraglutide was shown to prevent the loss of brain insulin receptors and synapses [74], and reversed cognitive impairments induced by Aβ oligomers in rodents and non-human primates [74,80]. Similarly, in animal models of chemical parkinsonism, liraglutide or exendin-4 treatment have been associated with increased neurogenesis, preservation of dopaminergic neurons and dopamine synthesis in *substantia nigra*, and the rescue of motor impairments [55,59,60,64]. The knowledge regarding the therapeutic benefits of GLP-1RAs in PD models based on the injections of aggregated alpha-synuclein is currently very limited. One study has reported that a long-acting GLP1-RA (NLY01) reduced de novo alpha-synuclein aggregation and dopaminergic neuronal loss in the striatum in a transgenic mouse of synucleinopathy, following the intra-striatal delivery of exogenous aggregated alpha-synuclein [81]. However, another study based on the delivery of aggregated alpha-synuclein in the olfactory bulb showed that a long-acting GLP1 analogue did not significantly reduced alpha-synuclein aggregation in the brain of wild-type mice [82]. In addition, there is a considerable amount of evidence demonstrating that GLP-1RAs reduce brain damage in animal models of ischaemic stroke and traumatic brain injury [38,39,54,61,63], and improve the survival of motor neurons in the mutant superoxide dismutase 1 (SOD1; G93A) model of motor neuron disease [50]. Nevertheless, a single unifying mechanism of action underlying the neuroprotective properties of GLP-1RAs or putative disease modification remains a matter of investigation [4,17,22,38,51,83].

The therapeutic utility of GLP-1RAs on clinical outcomes in neurodegenerative diseases is only recently beginning to emerge (Table 2). In a trial involving patients with impaired glucose tolerance or T2DM, liraglutide administration for 3 weeks improved short-term memory [84]. Although these findings are encouraging, clinical studies in patients with mild cognitive impairment (MCI) or AD have yet to conclude definite beneficial outcomes. For instance, liraglutide administration over 3 months in middle-aged individuals with cognitive complaints and/or a positive family history of AD increased connectivity in the default mode network, as detected by functional magnetic resonance imaging (fMRI) [85]. In a pilot study (NCT01469351, [86]) involving a small cohort of AD patients, 6 months’ treatment with liraglutide improved cerebral glucose uptake; however, no significant effects on Aβ plaque load or cognitive performance were noted [87]. In a phase II study involving patients with mild AD (ELAD, NCT01843075), 12 months’ treatment with liraglutide failed to improve cerebral glucose uptake, but there were encouraging indicators with regard to cognitive function and grey matter volume [88]. In addition, in an exploratory study involving AD patients, 18 months’ treatment with exenatide produced insignificant outcomes in cognitive measures, cortical thickness and volume, or biomarkers in biological fluids except for a slight reduction in Aβ42 within neuronal extracellular vesicles (NCT01255163) [89]. In the clinical trials involving PD patients (NCT01971242, NCT01174810), 12 months’ treatment with exenatide improved motor and cognitive outcomes [90,91], albeit, these findings have been described as ‘low certainty evidence’ [92]. In near future, semaglutide is planned to be tested in two large phase III studies involving MCI and AD subjects (EVOKE and EVOKE Plus; NCT04777396 and NCT04777409). Moreover, clinical trials using liraglutide, lixisenatide and semaglutide in PD treatment are also currently on the horizon (NCT02953665, NCT03439943, NCT03659682) [92]. 

Apart from the common animal models of neurodegeneration specified above, the neuroprotective effects of GLP1-RAs are also reported in some preclinical models of rare neurodegenerative diseases. Huntington’s disease is an autosomal dominant neurodegenerative disorder caused by the CAG repeat expansions in the huntingtin (*HTT*) gene, and clinically manifests as hyperkinetic movement disorder and cognitive decline. In a transgenic mouse model overexpressing mutant human huntingtin (N171-82Q), once-daily subcutaneous delivery of exendin-4 reduced huntingtin aggregates in the cortex, along with improvements in motor performance and overall survival [93]. Wolfram syndrome, caused by autosomal recessive inheritance of biallelic variations in the wolframin (*WFS1*) gene, is another rare neurodegenerative disorder that initially presents as diabetes mellitus. In a rat *Wfs1* knockout model, subcutaneous delivery of liraglutide over 6 months attenuated neuroinflammation and significantly reduced the loss of retinal ganglion cells and optic nerve axons [94]. However, we are not aware of any clinical trials with GLP-1RAs in Huntington’s disease or Wolfram syndrome.

## 3. Neurovascular Unit (NVU) in Ageing and Neurodegeneration

### 3.1. Neurovascular Coupling and Brain Energy Metabolism

The brain represents 2% of the body mass in adults, yet the energy demands to maintain adequate brain function require 20–25% of total body glucose utilisation and ~20% of the oxygen consumption [95,96]. In the adult brain, glucose is the obligate energy substrate for the production of adenosine triphosphate (ATP), except under particular circumstances where ketone bodies (e.g., during fasting) and lactate (e.g., during intense physical activity) are also utilised [97,98]. Circulating glucose enters the brain parenchyma by facilitated transport via 55-kDa glucose transporter 1 (GLUT1) on the capillary endothelial cells, and is subsequently taken up by neurons (via GLUT3) and glial cells [95,99,100]. The high basal metabolic rate in the brain is thought to be due to the energy demands for maintaining ionic gradients at the plasma membrane of neuronal processes by ATP-consuming pumps (chiefly Na^+^/K^+^-ATPase), as well as for the production and uptake of neurotransmitters [101,102]. Besides being the major substrate for ATP production, glucose is also required for the synthesis of neurotransmitters such as glutamate, γ-aminobutyric acid (GABA), acetylcholine and glycine [101]. 

The energy supply to the brain is tightly linked with neuronal activity and these phenomena form the basis of neurovascular coupling, and can be directly monitored by acquiring information about the regional brain glucose utilisation through imaging modalities such as fMRI and positron emission tomography (PET) [96,103,104]. Briefly, it is well established that task-dependent neural activity triggers an increase in the local cerebral blood flow (CBF) and glucose uptake in a pattern that is highly restricted to the activated network [105]. This stringent regulation is considered to be due to the lack of locally stored energy reserves that can be mobilised to meet the energy demands (except glycogen in astrocytes, elaborated below), as well as to remove by-products of neural activity (e.g., lactate, CO_2_, Aβ) and possibly for local temperature regulation [103,106,107,108,109]. The physiological basis of functional hyperaemia (see Glossary) is thought to reflect a vasoactive effect on the brain capillaries by the local accumulation of metabolic by-products during synaptic activity (e.g., lactate, CO_2_, H^+^) and/or the active generation of vasodilatory mediators (e.g., NO—nitric oxide, adenosine, ATP, K^+^) [106,110,111]. Moreover, glutamate released during synaptic activity raises intracellular calcium in astroglia through metabotropic mGluR5 receptors, with the subsequent stimulation of vasoactive arachidonic acid derivatives generation, which alters the tone of pericytes and vascular smooth muscle cells [99,106,112,113,114]. Other studies also implicate neuronal ATP release with a subsequent rise in intra-astroglial calcium through the activation of purinergic P2X or P2Y receptors [112,115,116]. 

The key basis of neurovascular coupling is the anatomical integration between the NVU and the tripartite synapse (i.e., pre- and postsynaptic elements, and astroglial processes) [95,99] (Figure 3A). The intraparenchymal brain capillaries contain a single layer of endothelial cells that are endowed with tight junctions, and are surrounded by a layer of pericytes and astroglial foot processes forming the glia limitans of the BBB. On the synaptic side, astroglial processes ensheath the synapses and are connected to each other via gap junctions. This anatomical arrangement has two implications: (i) the endothelial cells in the intraparenchymal brain capillaries provide a structural barrier for regulating the exchange of substances (e.g., nutrients, peptides, waste products) between blood-to-brain and brain-to-blood via specific efflux and influx transporters [117]; and (ii) the perivascular astrocytes detect the neural activity on the synaptic side (due to the presence of glutamate uptake transporters), while uptaking glucose from capillary blood by facilitated diffusion via 45-kDa GLUT1 transporter in their end feet [100]. Furthermore, the intimate proximity of astroglial end feet to the intraparenchymal brain microvessels is crucial for the adequate expression and proper localisation of transmembrane proteins forming the endothelial tight junctions [118]. With that overview in sight, it is apparent that CBF, glucose transport into brain parenchyma and glucose metabolism are tightly linked via the structural and functional features of the NVU [119]. Accordingly, pathological processes that lead to NVU disruption—either due to structural damage or dysregulation in the levels of local and/or circulating vasoactive factors—can cause brain dysfunction, and in the long-term lead to irreversible brain damage [110].

### 3.2. Structural Derangements of NVU in Ageing and Neurodegeneration

Ageing is a significant risk factor for several metabolic, cardiovascular and neurological conditions, including neurodegenerative diseases. Ageing affects cells and tissue both at the molecular and cytoarchitectural levels, such that there is a declining capacity to maintain healthy wear-and-tear, and the capacity to mount an appropriate cellular homeostatic response is also reduced [117,120,121]. The brain is particularly sensitive to the local and systemic effects of the age-related decline in cellular functions that may predispose to cognitive decline [121,122]. In this regard, the concept of ‘brain ageing’ is quite relevant, since it attempts to correlate the structural and functional alterations in the brain to the cognitive status of an individual [117,123]. In particular, the lack of cognitive and neurological symptoms in the elderly may not necessarily reflect the lack of pathological processes in the brain [122,124]. For instance, a comprehensive post-mortem assessment of 330 brains from cognitively normal elderly donors revealed that more than 50% of the brains exhibited markers of AD pathology, while as many as 40% showed evidence of mixed pathology (AD pathology with microvascular brain damage) or Lewy body pathology. Only about 4% of brains in this study showed lack of pathological markers associated with AD, PD or microvascular damage [121].

Several macroscopic, microscopic and molecular alterations in the brain parenchyma and supporting tissue (e.g., vasculature) hint towards defective energy extraction and/or utilisation during ageing that may adversely affect brain functions [117,125,126]. Neuroimaging studies in the elderly show that there are detectable changes in the volume of both grey and white matter, a reduction in the cortical thickness and/or abnormal functional connectivity in the absence of any severe neurological and/or cognitive disability [127,128,129,130,131]. However, rapid brain ageing in distinct regions (e.g., isocortex and/or hippocampus) is a prognostic indicator of developing cognitive impairment that can progress to overt dementia [132]. Similar observations have been reported with regard to the vulnerability of neuronal populations in substantia nigra to ageing-associated degeneration that may precede clinical PD [133]. In this regard, histological studies reveal that there is a mild to moderate decline in the number of pigmented and tyrosine hydroxylase-positive neurons (10–30% loss; 4–10% per decade) in as many as 1/3 of elderly subjects without clinically defined PD [134,135]. Moreover, it is also suggested that by the time of clinical presentation of PD, the loss of dopaminergic neurons in substantia nigra may exceed 50–70% [133,136]. Conventionally, the selective vulnerability of neuronal populations to disease processes is usually attributed to their inherent properties (i.e., high metabolic rate, ROS generation, and extensive arborisation with increased energy demand) [133,136,137]. However, it can also be argued that in susceptible individuals, distinct neuronal populations experience brain ageing at an accelerated rate, which leads to clinical presentation of a neurodegenerative disorder such as AD or PD.

Thus, during healthy brain ageing, individuals maintain a functional neuronal reserve for a considerably long time before the diagnosis of a neurological and/or cognitive disability is established [136,138]. Conversely, pathological factors such as protein aggregation and proteopathic stress, and possibly the influence of systemic comorbidities, accelerate brain ageing, leading to a decline in the neuronal reserve over time [132,139]. This is supported by epidemiological data suggesting that the dementia risk is greatly aggravated by several factors otherwise commonly associated with cerebrovascular disease, including hypertension, diabetes and obesity [140,141,142]. Accordingly, a number of MRI-based studies show that intraparenchymal microbleeds (see Glossary) are a feature observed during normal ageing. However, they are more frequent in individuals with MCI, early AD and in as many as 50% of dementia cases [143,144,145,146,147,148]. Furthermore, MCI and AD are associated with microbleeds in the infratentorial regions, a pattern of vascular damage also seen in individuals suffering from long-standing hypertension [106,143,149]. In addition, several microvascular alterations have long been associated with AD neuropathology, including increased atherosclerotic lesions, capillary tortuosity and rarefaction, thickening of the basement membrane and, in some cases, ‘string vessels’, which lack endothelial cells [103,150,151,152]. Similarly, post-mortem studies in long-standing PD cases reveal capillary damage and fragmentation in cortical areas and in brainstem regions, as well as increased endothelial cell nuclei in substantia nigra [153,154]. Moreover, a reduction in the neocortical CBF has been reported early in the course of PD, which correlates with cognitive impairment in some cohorts of PD patients [155,156,157].

### 3.3. Functional Derangements of NVU in Ageing and Neurodegeneration

It is plausible to postulate that accelerated brain ageing could be triggered by aberrations in the neurovascular coupling, thus exacerbating neurometabolic dysfunction in age-related neurodegenerative diseases [106,150]. First, with advancing age, CBF becomes uncoupled from glucose transport and utilisation both spatially and temporally [158,159,160,161,162]. Second, brain imaging studies that measure the metabolic rate of glucose utilisation using fluorodeoxyglucose (FDG-PET) indicate glucose hypometabolism in AD, PD and among the elderly at risk of AD [163,164,165,166]. For instance, decreased FDG uptake in the parietotemporal and cingulate cortex has been suggested to be quite specific for AD [167,168,169] or an indicator of enhanced risk for progression to dementia in individuals with MCI [170,171]. These features are reminiscent of structural and functional changes in the brains of middle-aged and elderly individuals with long-standing T2DM, with otherwise normal cognition, who also exhibit variable cortical atrophy and regional decline in the glucose metabolism.

Moreover, a growing body of evidence implicates brain insulin deficiency and/or insulin resistance as a key pathognomic phenomenon in both AD and PD, as indicated by the reduced expression of insulin (mRNA and protein), insulin receptors, decreased receptor affinity and reduction in the downstream signalling pathways (i.e., insulin receptor substrate 1, IRS1; phospho-AKT and phospho-glycogen synthase kinase 3β) [172,173,174,175,176,177,178,179]. Insulin is an important growth factor and is involved in dendritic sprouting, neuronal stem cell activation, cell repair and neuroprotection against Aβ [180,181]. In addition to the defects in central insulin signalling, it has also been suggested that systemic insulin resistance and hyperinsulinemia also impacts brain energy metabolism and memory [178,182]. In particular, T2DM has been described as a risk factor for AD and other neurodegenerative disorders [174,183,184]. For instance, studies show that systemic insulin resistance is associated with lower glucose metabolism in frontal, parietotemporal, and cingulate cortices [185,186]. In AD, decreased brain glucose uptake correlates with lower expression levels of the glucose transporters GLUT1 and neuronal GLUT3 [187,188,189], as well as with diminished activities of different metabolic enzymes such as glucose-6-phosphate isomerase, lactate dehydrogenase, phosphofruktinase, cytochrome oxidase, and α-ketoglutarate compared to the controls [173,178,190,191]. 

While the estimates on the prevalence of T2DM in AD are not unequivocally agreed upon, the prevalence of clinically diagnosed T2DM in PD has been reported to be as high as 8–30% and systemic insulin resistance in 50–80% subjects within different cohorts of PD patients [192,193]. Furthermore, T2DM in PD patients is markedly associated with the severity and progression of motor disability, cognitive decline and lower dopamine transporter binding in the striatum [194,195,196]. Also in PD, the dysregulation of different metabolic factors involved in brain glucose metabolism (e.g., glucose-6-phosphate dehydrogenase and 6-phosphogluconate dehydrogenase) has been reported to occur at early stages in the pathology [197]. Related to this, several loss-of-function mutations in genes known to be associated with early onset PD (e.g., *PARK2*) are directly linked to proteopathic stress and compromised mitochondrial functions [44]. Hence, it is plausible that systemic insulin resistance contributes to accelerated brain ageing and plays a role in the central energy crisis in the brain, which may be a harbinger of neurodegeneration in individuals at risk [182]. Further evidence of defective energy metabolism in AD and PD is also adduced by studies showing perturbations in the expression and/or activity of mRNA translation machinery required for adequate protein synthesis [198]. Since cellular protein synthesis is a highly energy consuming process, several of the factors involved in translation initiation and elongation phases are sensitive to declining ATP levels. Accordingly, post-mortem studies in AD and PD reveal considerable aberrations within the mammalian target of rapamycin (mTOR) pathway [199], AMP-activated protein kinase (AMPK) [200], elongation factor-2 kinase (eEF2K) [201,202], and stress-adaptive phosphorylation of the initiation and elongation factors [198,201,202,203].

## 4. GLP-1RAs in Neurodegeneration: The Neurovascular Connection

Given the pleiotropic indirect effects of GLP-1 signalling on cardiovascular function, blood glucose homeostasis, fatty acid metabolism and cell growth, the mechanism(s) of GLP-1RAs mediated neuroprotection warrant a careful assessment. As alluded to earlier, the effects of GLP-1RAs have been studied in animal studies of neuronal damage involving acute CNS injury (stroke, traumatic brain injury), sub-acute toxin induced dopaminergic loss and prolonged effects of proteopathic stress (e.g., transgenic models of Aβ amyloidosis/tau aggregation) [22,38]. A corollary to the neurocentric mode of action (i.e., direct neuroprotection mediated by GLP-1 receptor signalling in neurons or modulation of Aβ/tau aggregation) would be the ancillary mechanism of improved neurovascular coupling. Accordingly, GLP-1 receptor signalling and/or GLP-1RAs could promote a favourable intercellular milieu at the level of NVU, which in turn is associated with a broad range of beneficial effects in the CNS. For instance, this is supported by observations that exendin-4 treatment in a mouse of atherosclerosis (western-type diet) with human-like lipoprotein metabolism was associated with reduced inflammation, and macrophage influx in the vessel wall and in the liver [204]. Similarly, intraperitoneal delivery of liraglutide in a mouse model of polymicrobial induced sepsis attenuated endothelial dysfunction, vascular inflammation and oxidative stress in the aortic wall [205]. 

To illustrate this point further, we provide here some examples with the caveat that our understanding of specific cellular effects of long-term GLP-1RAs on the components of NVU is still incomplete. The existing literature supports the view that the predominant direct effect of endogenous GLP-1 receptor signalling and pharmacological GLP-1RAs is to promote improved glycaemic control, reduce systemic insulin resistance and decrease free fatty acids in circulation [1,3,4]. Systemic insulin resistance and consequent dysregulation in homeostatic glycaemic status are the key mediators of the chronic complications associated with uncontrolled T2DM and metabolic syndrome, both in the form of microvascular disease (retinopathy, neuropathy, nephropathy) and macrovascular complications (stroke and coronary artery disease) [206,207]. The underlying metabolic driver of the microvascular complications in prediabetes and T2DM is uncontrolled glucose entry into the cells, which leads to overproduction of mitochondria-derived reactive oxygen species (ROS) due to an increased flux of glucose through the tricarboxylic acid (TCA) cycle. This has directly been implicated in endothelial cell dysfunction and creating a pro-inflammatory milieu, with damage to the vessel wall and normal haemodynamic response [206,207]. For instance, a pathologic factor associated with vascular dysfunction and inflammation is the accumulation of irreversibly glycated proteins (i.e., AGEs—advanced glycosylation end-products). AGEs are known to directly alter endothelial gene expression through transcriptional repression, as well as to activate macrophages by binding to the AGE receptors (RAGEs) and drive downstream inflammatory gene expression involving nuclear factor kappa B (NF-κB) [208]. In the context of chronic disturbances of glycaemic indices, some studies show that the AGEs accumulation is more frequently encountered in AD brain than in age-matched controls [209,210]. AGEs have been detected in association with Aβ plaques, as well as with neurofibrillary tangles, both in neurons and glial cells [211]. These reports also suggest that AGEs stimulate tau hyperphosphorylation in vivo, and the glycation of Aβ influences its aggregation [209,211,212]. In addition to the formation of AGEs, intracellular hyperglycaemia leads to the increased formation of diacylglycerol (DAG) from the glycolytic intermediate glyceraldehydes-3-phosphate, which in turn is a potent stimulus for protein kinase C (PKC). The inappropriate activation of PKC affects blood flow and vascular permeability in part via the local synthesis of NO [206,207].

As mentioned above, deciphering the significance of systemic insulin resistance and adequate glycaemic control in neurodegenerative diseases remains an evolving field. For instance, a recent study showed that AD patients exhibit a reduced peripheral insulin sensitivity and hyperinsulinemia, both under fasting state as well as in response to an oral glucose tolerance test [213]. Chronic peripheral hyperinsulinemia is expected to cause the downregulation of insulin receptors at the NVU and reduction in the insulin entry into brain, as reflected in lower brain insulin concentration in AD patients [214,215]. As a consequence, this would lead to aberrations in the glucose transport at the NVU, as reflected by the FDG-PET studies, as well as the reduced expression of GLUT1 in the brain capillaries of AD patients [216]. The significance of glucose transport at the NVU in relation to protein aggregation in the brain has not been extensively studied to date. Nevertheless, studies show that haploinsufficiency of GLUT1 in a transgenic mouse model of Aβ amyloidosis leads to microvascular degeneration and CBF reduction that precedes progressive neuronal loss and intracerebral Aβ deposition [217]. 

## 5. Unresolved Questions and the Need for Novel Experimental Paradigms

Translating the pleiotropic effects of GLP-1 receptor activation into the neuroprotective properties ascribed to GLP-RAs represents an interesting scientific quest. In addition to the neurocentric context highlighted earlier, it is apparent that novel experimental paradigms are needed to fully establish the therapeutic relevance of GLP-1RAs in neurodegenerative diseases. Taking into consideration the structural and functional perturbations of the NVU in neurodegenerative disorders, some of the approaches could address the direct and indirect effects of GLP-RAs on the physiology of NVU. For instance: (i) probing the effects of GLP-1RAs on the cellular components of the NVU, in particular the metabolic and trophic support activity of astroglia in the context of neurodegenerative proteinopathies and (ii) investigating the relevance of GLP-1RAs activity in the periphery to neuroprotective outcomes in brain pathologies. 

### 5.1. Effects of GLP1-RAs at the NVU

With regard to the first aspect, there is burgeoning evidence that GLP-1 signalling is involved in the modulation of astroglial metabolism and their response to CNS injury. For example, transient cerebral ischaemia in gerbils significantly increased the immunoreactivity of GLP-1 receptors in the pyramidal neurons of the hippocampal CA1 region within the first 2 days after the injury [218]. Intriguingly, this acute phase was followed by a rapid decline in the neuronal GLP-1 receptor expression, and shifted to increasing GLP-1 receptor expression by GFAP-positive astrocytes and some GABAergic interneurons in the CA1 over 4–10 days post-injury [218]. In the same report, the authors showed that exendin-4 administration (2 h prior to surgery and 1 h after reperfusion) prevented neuronal cell loss and infiltration by activated microglia. While the significance of delayed upregulation in astroglial GLP-1R expression remains unsettled, it possibly represents an injury-specific response that may affect the neuron–glia interaction after ischaemia and reperfusion [218]. Furthermore, a long-acting brain penetrant, GLP-1RA (NLY01), prevents the conversion of astrocytes into neurotoxic A1 phenotype in a mouse model of synucleinopathy [81]. Recent evidence also suggests that GLP-1 receptor signalling in astrocytes may also be involved in the neuron–astroglia metabolic coupling, both under physiological and pathological states. Astroglia are integral players in the regulation of brain energy metabolism, and crucial players in the cellular phase of neurodegeneration in AD [48,113]. With their end feet in contact with the capillary endothelial cells, astrocytes are the first cellular barrier encountered by the circulating glucose subsequent to its entry into brain parenchyma [95]. Immunohistochemical analyses show that astroglial processes form closely apposed contacts with GLP-1 receptor-positive axonal processes in the NTS [219]. The same study also demonstrated that exendin-4 increased calcium signalling in astrocytes ex vivo, and localised to NTS astrocytes following in vivo administration [219]. Intriguingly, mice lacking GLP-1 receptors in the glial fibrillary acidic protein (GFAP)-positive astrocytes exhibited increased cerebral glucose uptake, improved memory formation and spontaneous activity in the midbrain dopaminergic neurons [220]. The same report also showed that the genetic deletion of the GLP-1 receptor in astrocytes disrupted mitochondrial integrity, which was associated with the activation of an integrated stress response involving fibroblast growth factor-21 [220]. 

It is also noteworthy that neurons and astrocytes differ in the mode of glucose utilisation for ATP generation. In neurons, glucose is processed through ROS-generating oxidative phosphorylation in mitochondria, while astrocytes preferentially utilise glucose via aerobic glycolysis-generating pyruvate and lactate [221,222]. At the glutamatergic synapses (the most common excitatory synapses in the CNS), the neuron–astroglia metabolic coupling during synaptic activation involves neuronal glutamate release and rapid uptake by surrounding astroglia. This triggers aerobic glycolysis and lactate release by the astrocytes, and the released lactate is considered a fuel for ATP generation within neurons—a process commonly referred to as the Astrocyte Neuron Lactate Shuttle (ANLS) [95,223,224] (Figure 3B). In the neurophysiological context, lactate has been implicated in regulating neuronal gene expression for survival, synaptic plasticity and in memory formation [98,109,225,226]. In a recent study, it was shown that liraglutide increased aerobic glycolysis and lactate release, with a concomitant reduction in oxidative phosphorylation by cultured astrocytes. In neuron–astroglia co-cultures, liraglutide also reduced Aβ neurotoxicity [69]. However, another study showed that liraglutide inhibited glucose uptake in cultured astrocytes, while promoting β-oxidation of fatty acids for ATP generation [220]. Further related to the glucose metabolism, another feature that is peculiar to astroglial cells (although not exclusive) is the glycogen storage, which can be mobilised in the events of increased energy demand and during brain repair [97,221]. The evidence for this is based on the observations that glycogen levels in the brain are increased when neuronal activity is inhibited (e.g., under general anaesthesia) [227]. Moreover, reactive astrocytes in the vicinity of damaged neuropil exhibit the increased accumulation of glycogen [228], which is interesting in view of the upregulation in astroglial GLP-1 receptor expression following ischaemic injury, as mentioned above [218]. While we are not aware of any study to date that has directly investigated the effects of GLP-1RAs on astroglial glycogen storage and/or turnover, studies in human and rodent skeletal muscle and hepatocytes indicate that GLP-1 receptor signalling enhances glycogen synthesis [229,230]. 

Apart from the metabolic aspects, the glutamate-scavenging function of astroglia is vital to maintain neuronal excitability and to prevent glutamate-induced excitotoxic cell death. Glutamate excitotoxicity is a crucial mediator of neuronal death in ischaemic stroke, and is also widely implicated in some neurodegenerative disorders [48,231]. In this context, the administration of GLP-1RAs (e.g., exendin-4 and liraglutide) prior to or following cerebral ischaemia in animal models of stroke have consistently been reported to reduce infarct size, mitigate oxidative stress and improve endothelial function [38,39,54,61,63]. The neuroinflammatory cascade following the excitotoxic insults within the CNS also affect GABA receptor-mediated inhibitory signalling by eliminating GABAergic synapses, thus exacerbating the imbalance in the neuronal excitation/inhibition and possibly contributing to the NVU dysfunction (i.e., BBB disruption) [232]. Intriguingly, pancreatic β-cells are among the extra-neural sites expressing high levels of glutamate decarboxylase, an enzyme that catalyses the formation of GABA from glutamate [233,234]. Under hyperglycaemic conditions, GLP-1 inhibits glucagon secretion by α-cells through paracrine mechanisms that involve β-cell-derived GABA and insulin [235,236,237]. While the details of these mechanisms are not fully understood, it is considered that insulin increases the membrane translocation of GABA-A receptors in α-cells, which are activated by the GABA derived from the neighbouring β-cells. This leads to membrane hyperpolarisation and a consequent reduction in the glucagon secretion by the α-cells. Moreover, in cultures of rat hippocampal neurons, insulin stimulation increases the density of postsynaptic GABA-A receptors, presumably mediated by phosphorylated Akt (Thr308) [238]. The interplay of GLP-1 receptor activation and GABA-mediated synaptic effects has largely been studied in the brain centres regulating food intake, energy consumption and glucose disposal [239,240]. Hence, a detailed characterisation of the neuroprotective actions attributed to GLP-1RA in models of excitotoxic neuronal loss should be carefully evaluated to incorporate excitation/inhibition balance, and possibly in a region-specific manner. 

### 5.2. Effects of GLP1-RAs in Periphery and Inter-Organ Communication

In addition to the direct effects on the cellular elements of the NVU, ancillary mechanisms of neuroprotection by GLP-1RAs may indirectly emanate from the attenuation of pathological processes in the periphery, in particular, favourable outcomes associated with GLP-1RAs on glucose and lipid metabolism, cardiovascular function and systemic inflammation [3,4,241]. As discussed above, the existing literature in animal models of Aβ amyloidosis and tauopathies indicates that the long-term activation of GLP-1 signalling attenuates protein aggregation in the brain (Table 2). Nevertheless, these effects are considered to be the direct result of GLP-1RAs in the CNS and the systemic factors are not usually taken into consideration. Although peripherally administered GLP-1RAs are widely considered to be BBB permeable [3,242], some studies also hint that GLP-1 receptor signalling in the periphery influences central GLP-1 receptors and vice versa; however, the existing literature on the brain-periphery GLP-1 axis is largely limited to the control of food intake and blood glucose disposal [20,37]. It also implies that peripheral actions of GLP-1RAs, in particular reduced systemic inflammation and oxidative stress, may indirectly mitigate pathogenic processes involved in neurodegeneration. For instance, AD is associated with elevated levels of several pro-inflammatory cytokines in circulation (e.g., interleukin 6, IL-6; IL-12, tumour necrosis factor-alpha, TNF-α), that may impact the NVU function [243,244,245]. This is also supported by epidemiological and neuropathological observations that indicate that there is a considerable overlap between cerebrovascular pathology and the risk of developing dementia [246]. 

In a broader context with regard to the peripheral effects of GLP1-RAs, aberrations in the physiological inter-organ communication represents an unexplored area in neurological diseases, especially AD and PD. In extreme scenarios, brain function is markedly affected by the health of other organs, as illustrated by the presence of hepatic encephalopathy in patients with cirrhosis [247], and uremic encephalopathy in patients with declining kidney function [248]. In this context, liver or kidney dysfunction has been suggested to increase the risk for AD and PD [249,250,251,252,253]. While the global burden of age-related neurodegenerative disorders is on the rise, this association is especially worrying due to the increasing global prevalence of chronic liver diseases, mainly non-alcoholic fatty liver disease (NAFLD) as well as chronic kidney disease [254,255]. The pathophysiological mechanisms linking these disorders are not well understood; however, it is likely that shared mechanisms such as insulin resistance, inflammation and endothelial dysfunction may play a role. In addition, the pathogenic significance of a reduction in the hepatic/renal clearance and subsequent chronic accumulation of neurotoxic metabolites, either as a result of ageing and/or comorbidity in AD and PD, is largely unknown. For example, polymorphisms in the genes encoding *CYP1A1* and *CYP2D6,* members of the cytochrome P450 family and involved in detoxification reactions, increase the risk of PD [256,257]. Conversely, neuroinflammatory changes in the CNS can also affect peripheral organ function; for instance, damage to the dopaminergic neurons in substantia nigra alters the expression of hepatic P450 enzymes [258]. Thus, it is worthwhile examining whether the putative neuroprotective effects of GLP1-RAs may partly emanate from improving and/or stabilising the functionality of the liver and/or kidney, i.e., excretory and detoxifying functions of the body. Supporting this, Mantovani and colleagues performed a meta-analysis of 11 randomised controlled trials (involving a total of 936 middle-aged individuals) using GLP-1RAs for the treatment of NAFLD or non-alcoholic steatohepatitis (NASH). Their analysis revealed that treatment with GLP-1RAs for a median of 26 weeks was associated with significant reductions in steatosis and serum liver enzyme levels, as well as with greater histological resolution of NASH [259]. However, it remains unclear whether these beneficial effects are due to a direct effect on the liver, since there is still a debate on whether the GLP-1 receptors are expressed in hepatocytes [260,261,262]. Nevertheless, there are indications that GLP-1RAs may have therapeutic benefit in the management of patients with NAFLD [263]. 

Regarding GLP-1RAs and diabetic kidney disease, beneficial effects on composite renal outcomes have been reported, mainly the prevention of new-onset macro-albuminuria, albeit more evidence is needed to establish therapeutic benefits in clinical settings [2,264]. Hence, the ongoing FLOW trial (NCT03819153) is assessing the effects of semaglutide on complex renal outcomes in patients with diabetic kidney disease [265]. Further evidence regarding the therapeutic utility of GLP-1RAs in improving inter-organ communication is hinted by the modulatory effects of GLP-1 receptor activation on the renin-angiotensin–aldosterone system (RAAS) [266]. The RAAS is one of the key mechanisms in the long-term regulation of arterial blood pressure through its actions on the vascular smooth muscle tone and renal reabsorption of sodium and water. Perturbations in the RAAS are implicated in the progression of cardiovascular dysfunction and nephropathy associated with poorly controlled T2DM [266]. Biochemical studies also suggest that in addition to the peripheral RAAS axis, the brain harbours a local RAAS system with the expression of several components in neurons and glia [267]. While the detailed discussion of RAAS regulation in health and disease is beyond the scope of this article, existing reports implicate RAAS activity in memory and cognition, as well as in the pathophysiology of AD and PD [267,268]. A direct link between GLP-1RAs and RAAS is adduced by observations that liraglutide increased the expression of angiotensin-converting enzyme 2 (ACE2) and promoted vasodilation in vivo, presumably via increased activation of the Mas receptors and NO synthesis [269,270]. In this regard, it is worth mentioning that a limited number of studies have shown that the activation of Mas receptors with angiotensin (1–7) ameliorates phenotypic defects in the experimental models of ischaemic stroke, 6-OHDA-induced parkinsonism and Aβ amyloidosis [271,272,273]. Based on these reports, it remains to be investigated whether GLP-1RAs mitigate cognitive impairment and brain ageing, in part, by maintaining/restoring healthy inter-organ communication.

Another pertinent area for investigation would be the effects of GLP1-RAs on the gut microbiota, since alterations in the composition of gut microbiota or their metabolic products are reported in AD and PD cohorts [274,275]. This is relevant, since systemic metabolic disorders including T2DM affect the composition of gut microbiota [276,277], potentially leading to a milieu that is conducive to pathological processes relevant to neurodegeneration [274,275]. This aspect is also relevant, since there is an emerging notion that some CNS proteinopathies might originate in the periphery. For instance, a number of post-mortem neuropathological and experimental observations suggest that alpha-synuclein aggregation may originate in the periphery (e.g., gut, peripheral nerve afferents) and invade the nervous system in a manner reminiscent of prions [278,279,280]. However, refined experimental models are needed to fully understand the interaction between peripheral GLP-1 signalling and protein aggregation in the CNS. One potential approach to investigate the peripheral effects of GLP-1RAs could be to employ experimental models of CNS proteinopathies in which protein aggregation is induced in the periphery, for instance by injecting aggregated alpha-synuclein into the gut or propagation involving neuromuscular junction [280,281,282,283]. In these models of synucleinopathy, the peripheral induction of alpha-synuclein aggregation is associated with variable degrees of neuronal loss and neuroinflammation in the CNS. Furthermore, the peripheral-to-central propagation of aggregated alpha-synuclein in a transgenic mouse model leads to mechanical allodynia and impaired nociception [280]. Interestingly, enhancing GLP-1 by teneligliptin (DPP-4 inhibitor) in a rat model of partial sciatic nerve transaction ameliorated neuropathic pain and reduced inflammation in the spinal cord [284]. Similar experimental paradigms would be valuable to further elucidate the mechanism(s) that may underlie neuroprotection associated with GLP-1RAs, i.e., investigating whether GLP-1RAs directly affect protein aggregation via peripheral anti-inflammatory effects and/or prevent pathological processes in the CNS predominantly through the central effects. 

## 6. Conclusions

Since the publication of reports showing the involvement of GLP-1 receptor signalling in cognitive function, and encouraging outcomes with GLP-1RAs in experimental models of acute or chronic neuronal toxicity, there is an engaging discussion regarding the mechanism(s) that underlie their neuroprotective mode of action. In addition to their direct pharmacological actions involving GLP-1 signalling and consequent promotion of glycaemic control, a range of indirect effects associated with the long-term use of GLP-1RAs signify tissue-specific mechanisms with translational potential for GLP-1-based therapies (Table 1). We expect that understanding the pharmacological actions of long-term GLP-1RAs use on the structural and functional aspects of the NVU, in particular neuron–astroglia metabolic coupling, will aid in filling critical gaps in the knowledge regarding the therapeutic utility of these agents in neurological conditions. With relevant examples, we have also outlined some experimental paradigms that could be further explored for studying the direct and indirect effects of GLP-1RAs in neurodegenerative diseases that could help elucidate putative disease-modifying effects.

While the outcomes from the clinical trials regarding the therapeutic potential of GLP-1RAs (and DPP-4 inhibitors) in AD and PD are eagerly anticipated in the near future (Table 2), some additional considerations are worth pointing out for the transition into the clinical setting. First, guidelines for patient stratification may be required to identify potential responders and non/weak responders, for example due to inter-individual differences in GLP-1 receptor expression or polymorphisms that may affect response to GLP-1RAs, or clinical evidence indicating advanced stages of neurological dysfunction [2]. Second, it needs to be established whether enhancing GLP-1 receptor signalling would result in clinical improvements via direct central effects, and/or indirectly via promoting healthy brain ageing through a concerted interplay of brain and peripheral organ systems. Hence, a comprehensive framework may be required that incorporates standardised measures of neurological/cognitive functions, as well as markers of metabolic status (e.g., glycaemic control) and co-morbidities (e.g., hypertension, obesity, liver or kidney disease). Third, meticulous post-mortem neuropathological assessments of volunteer donors from these trials would be needed to establish the evidence for disease modification (i.e., reduction in protein aggregation and neuronal loss), as suggested by the animal studies (Table 2). Lastly, refinements in brain imaging modalities may provide additional clues to pinpoint the actions of GLP-1 receptor activation on the NVU. Although this sounds technically challenging, there are examples for assessing NVU function and integrity via fMRI using the blood oxygen level-dependent signal [285], representing potential areas to guide such refinements. 

## Figures and Tables

**Figure 1 cells-11-02023-f001:**
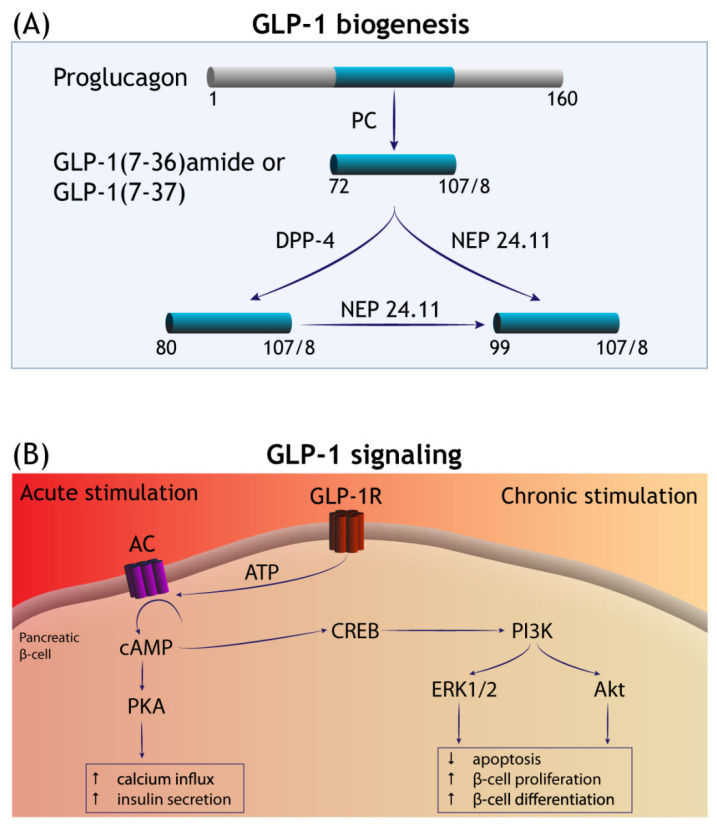
GLP-1 biogenesis and signalling transduction. (**A**) GLP-1 is liberated from the precursor molecule proglucagon—in the enteroendocrine L cells and in the CNS—by the isoforms of prohormone convertase (PC). GLP-1 (7–36)-amide and GLP-1 (7–37) are the major bioactive forms in circulation and degraded by the serine protease dipeptidyl peptidase (DPP-4) and neutral endopeptidase 24.11 (NEP 24.11). (**B**) The acute phase of GLP-1 receptor (GLP-1R) stimulation involves the activation of adenylate cyclase (AC) and a rise in the levels of intracellular cyclic (AMP), with subsequent engagement of additional effector mechanisms mediated by protein kinase A (PKA). This culminates in increased calcium influx and the promotion of insulin secretion by the islet β-cells. Long-term (chronic) stimulation of GLP-1R is thought to occur via cAMP-responsive element binding (CREB) signalling that promotes pro-survival gene expression mediated by the receptor tyrosine kinases PI3K/Akt and extracellular signal-regulated kinase (ERK)1/2, among others (see Introduction). In the pancreas, this increases β-cell mass by diminishing apoptosis and increasing β-cell proliferation and differentiation.

**Figure 2 cells-11-02023-f002:**
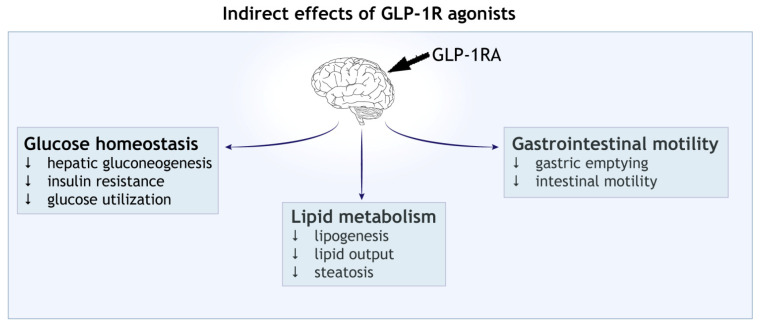
Indirect effect of GLP-1 receptor agonists (GLP-1RAs). Activation of central GLP-1 receptor signalling in the brain leads to metabolic reprogramming in the peripheral tissues. Shown are the effects on blood glucose homeostasis, lipid metabolism and gastrointestinal motility.

**Figure 3 cells-11-02023-f003:**
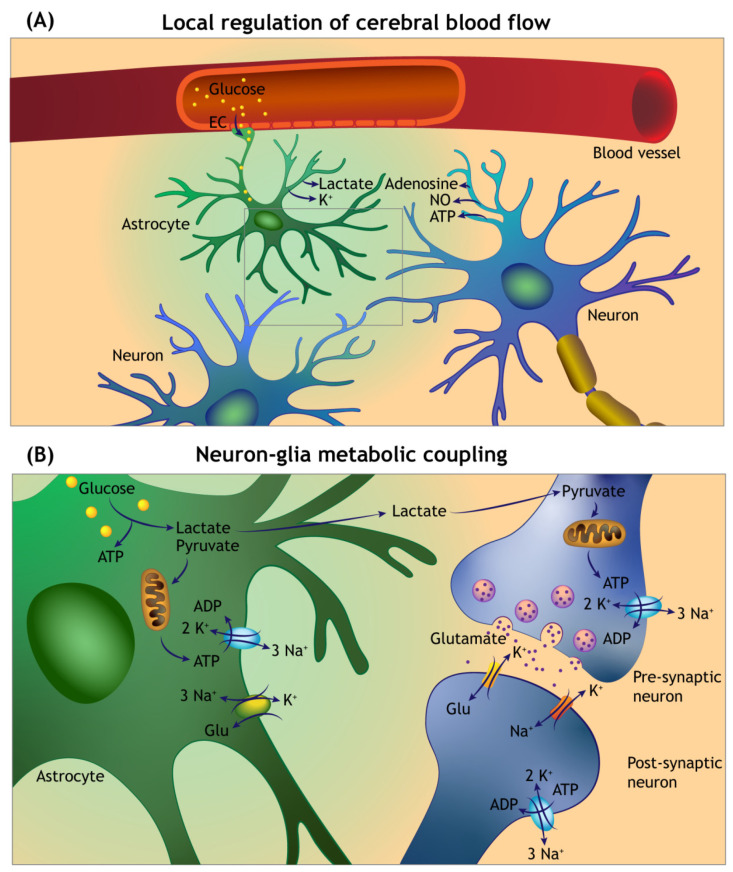
Overview of the local cerebral blood flow auto-regulation and neuron–glia metabolic coupling. (**A**) Neuronal and astroglia-derived vasoactive mediators act on the intraparenchymal capillaries and arterioles to sustain enhanced delivery of metabolic fuels during functional hyperaemia. Circulating glucose enters into the brain parenchyma by facilitated diffusion via glucose transporters in the capillary endothelial cells (EC), and astroglial foot processes forming the glia limitans of the blood–brain barrier. (**B**) Glutamate uptake at the tripartite synapse by astrocytes is accompanied by sodium (Na^+^) entry, which triggers anaerobic glycolysis and lactate release. The lactate is transferred to the neurons (via a process termed the Astrocyte Neuron Lactate Shuttle) and converted to pyruvate by neuronal lactate dehydrogenase (not shown). The pyruvate enters the citric acid cycle in mitochondria for ATP generation. A significant amount of the ATP generated is used to maintain the function of sodium–potassium pumps (Na⁺/K⁺-ATPase).

**Table 1 cells-11-02023-t001:** GLP-1RAs: backbone modifications, frequency of administration and half-life.

GLP-1RA	Backbone Modification	Frequency of Administration	Half-Life
Exenatide (Byetta/Bydureon)	Exendin-4 (resistant to DPP-4 cleavage, largely due to the substitution of the second amino acid from alanine to glycine)	Twice daily/weekly	3.3–4 h
Lixisenatide (Adlyxin, Lyxumia)	Non-acylated GLP-1 (7–37) analogue based on exendin-4, but is modified by the deletion of one proline residue and with a C-terminal hexa-lysine extension	Daily	2.6 h
Oral Semaglutide (Rybelsus); Semaglutide (Ozempic)	Acylated Human GLP-1 (7–37) analogue	Once daily; weekly	1 week
Liraglutide (Victoza)	Mammalian GLP-1, substitution of lysine for arginine at position 28 with the addition of C-16 fatty acid	Daily	13 h
Dulaglutide (Trulicity)	Mammalian GLP-1; the GLP-1 portion of the molecule is fused to an IgG4 molecule, limiting renal clearance and prolonging activity	Weekly	1 week
Albiglutide (Eperzan and Tanzeum)	Two GLP-1 (7–36) molecules fused in tandem to human serum albumin	Weekly	1 week
Taspoglutide	Modifications designed to delay DPP-4 cleavage and other serine proteases, with greater receptor binding	Weekly	1 week

**Table 2 cells-11-02023-t002:** GLP-1RAs in AD and PD: An overview of the experimental models and clinical trials.

Studies	Experiment	GLP-1RA	Observations	Publications
** *Preclinical studies* **	** *Animal model* **			
** *AD features* **				
*Plaque load*	APP/PS1/tau mice 5xFAD mice APP/PS1 mice 3xTg-AD mice	Liraglutide Liraglutide Lixisenatide Exendin-4	Reduction of plaque load Reduction of plaque load Reduction of plaque load Reduction of plaque load	[72,76,77] [78] [76] [62]
*Tau phosphorylation*	APP/PS1/tau mice hTauP301L mice Aβ injection in mice APP/PS1 x db/db mice Streptozotocin injection in mice	Liraglutide Liraglutide Liraglutide Liraglutide Dulaglutide	Reduction of neurofibrillary tangles Reduced Tau phosphorylation Reduced Tau phosphorylation Reduced Tau phosphorylation Reduced Tau phosphorylation	[56,72] [58] [67] [79] [73]
*Cognitive and* *memory performance*	Aβ injection in mice Aβ injection in rats Streptozotocin injection in mice	Liraglutide Lixisenatide Dulaglutide	Improved cognitive impairment Improved spatial memory Improved memory ability	[74] [80] [73]
*Other*	Aβ injection in non-human primates	Liraglutide	Reduced synaptic loss	[74]
** *PD features* **				
*Dopaminergic neuronal loss*	6-OHDA rat model 6-OHDA rat model 6-OHDA rat model	Liraglutide Exendin-4 Exendin-4′	No influence on dopaminergic neuronal loss Neurogenesis Reduced lesions	[59] [55] [60]
*Motor performance*	MPTP mouse model MPTP mouse model	Liraglutide Lixisenatide	Improved motor control Improved motor control	[64] [64]
*α-synuclein aggregation*	Preformed fibrils injection in striatum of human A53T α-synuclein mice Preformed fibrils injection in the olfactory bulb of C57BL/6J mice	Exendin-4 (NLY01) Exendin-4	Reduced loss of dopaminergic neurons and improved motor performance No significant reduction of α-synuclein aggregation	[81] [82]
** *Clinical trials* **	** *Trial ID* **			
** *AD* **	NCT02140983	Liraglutide	Increased connectivity in the default mode network	[85]
	NCT01469351 NCT01843075 NCT01255163 NCT04777396 NCT04777409	Liraglutide Liraglutide Exenatide Semaglutide Semaglutide	Improved cerebral glucose uptake Improved cognition No significant changes in cognition Recruiting Recruiting	[86,87] [88] [89]
** *PD* **	NCT01971242 NCT01174810 NCT02953665 NCT03439943 NCT04154072 NCT03659682	Exenatide Exenatide Liraglutide Lixisenatide Exenatide Semaglutide	Improved motor and cognitive outcomes Improved motor and cognitive outcomes Active Active Active Not yet recruiting	[90] [91]

## Data Availability

All of the data cited in the article can be accessed in the original research studies provided in the reference list.

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
