# Peer review of "GLP-1 Receptor Agonists in Neurodegeneration: Neurovascular Unit in the Spotlight"

_cells, 2022, doi:10.3390/cells11132023_

Round 1
Reviewer 1 Report
Reviewer report
Giulia Monti review named “GLP-1 receptor agonists in neurodegeneration: Neurovascular unit in the spotlight” is discussing the possibilities (drug repurposing) to use well known drugs meant to treat diabetes against Alzheimer disease (AD) and Parkinson disease (PD). More specifically, the work is discussing the possibilities to use glucagon-like peptide 1 (GLP-1) receptor agonist to treat AD and PD as those drugs have been shown excellent neuroprotective effects in several in vitro and in vivo models.
In really simplified model, during the meal intestinal L cells are releasing GLP1 to the blood stream what activates GLP1r in pancreatic beta cells. This activation “says” to the beta cells that sugar is coming and beta cells starts to prepare to produce and release insulin. On the other hand, GLP1 enters to the brain and to”say” the body do not eat any more. Anyway, on the background the activation of GLP1r activates multiple cellular events: some examples, not limited, it decreases ER stress, improves protein synthesis, mitochondrial function and autophagy, reduces inflammation, ROS accumulation and even improves Iron homeostasis. All this results better beta cell “health” and improves insulin synthesis and release. As GLP1r is widely expressed in the body (like lungs, heart and CNS) then the therapeutic effects of its agonists are indicated in other disease models like cardiovascular and lung diseases, stroke and in many neurodegenerative diseases including but not limited PD, AD, Amyotrophic lateral sclerosis (ALS) and Wolfram Syndrome. Probably therapeutics mechanisms are the same or similar. It would be not bad to mention the fact that GLP1r agonists have been shown to have strong effects to delay the neurodegeneration in some rear disease animal models. This will defeatedly improve literature overview part.
Minor comments:
As I mentioned, the activation of GLP1r results multible cellular events and most of them are discussed including that GLP1r is going up in reactive glia during the injury. Actually, it goes up shortly after injury, then it goes down for few days and starts to go up again after 6-10 days after injury, during the repair period. Might be this phenomenon should be better explained and discussed? Indeed, on the same experiment they showed that activation GLP1r by its agonist reduced remarkedly reactive glia and improved brain regeneration.
More, GLP1r agonist have been shown to act as a modulators of renin angiotensin aldosterone (RAAS) pathway. Specifically, they have been shown to improve ACE2 activity, resulting the activation of compensatory axis of RAAS (Agtr2, Mas1 axis), what also improves cellular regeneration or apoptosis, reduces inflammation and fibrosis and makes blood vessels more permeable for minerals and nutrients.
On the other hand, might be the connection of GABA and GLP1 should be discussed. It seems that the activation of GLP1r increases GABA synthesis both in beta cells and in GABAeric neurons. In the pancreatic beta cells, GABA is released during insulin secretion. Insulin is increasing alpha cell GABA receptor expression and after GABA binds of it, glucagon secretion is reduced resulting the block of gluconeogenesis. On the same time GABA can bind to pro-alpha cells to induce their proliferation to beta cells. And it seems, that GABA needs GLP1r to exert its pancreatic function. Same in CNS, there have been shown that the activation of GLP1r in GABAeric neurons improves neuronal plasticity. Might be this should be also discussed.
thank you!
Author Response
RESPONSES TO REVIEWER 1 COMMENTS:
We appreciate the time and effort of the reviewer in suggesting improvements to the manuscript. The following is a point-by-point response to the reviewer’s comments.
- New text in relevant sections is highlighted in green
REVIEWER 1
REVIEWER COMMENT. Giulia Monti review named “GLP-1 receptor agonists in neurodegeneration: Neurovascular unit in the spotlight” is discussing the possibilities (drug repurposing) to use well known drugs meant to treat diabetes against Alzheimer disease (AD) and Parkinson disease (PD). More specifically, the work is discussing the possibilities to use glucagon-like peptide 1 (GLP-1) receptor agonist to treat AD and PD as those drugs have been shown excellent neuroprotective effects in several in vitro and in vivo models.
In really simplified model, during the meal intestinal L cells are releasing GLP1 to the blood stream what activates GLP1r in pancreatic beta cells. This activation “says” to the beta cells that sugar is coming and beta cells starts to prepare to produce and release insulin. On the other hand, GLP1 enters to the brain and to”say” the body do not eat any more. Anyway, on the background the activation of GLP1r activates multiple cellular events: some examples, not limited, it decreases ER stress, improves protein synthesis, mitochondrial function and autophagy, reduces inflammation, ROS accumulation and even improves Iron homeostasis. All this results better beta cell “health” and improves insulin synthesis and release. As GLP1r is widely expressed in the body (like lungs, heart and CNS) then the therapeutic effects of its agonists are indicated in other disease models like cardiovascular and lung diseases, stroke and in many neurodegenerative diseases including but not limited PD, AD, Amyotrophic lateral sclerosis (ALS) and Wolfram Syndrome. Probably therapeutics mechanisms are the same or similar. It would be not bad to mention the fact that GLP1r agonists have been shown to have strong effects to delay the neurodegeneration in some rear disease animal models. This will defeatedly improve literature overview part.
AUTHOR RESPONSE: Thank you for the overall constructive comments.
We have now added the studies on rare neurodegenerative diseases: Huntington disease and Wolfram syndrome, please see the highlighted text on Page 8.
Minor comments:
REVIEWER COMMENT. As I mentioned, the activation of GLP1r results multible cellular events and most of them are discussed including that GLP1r is going up in reactive glia during the injury. Actually, it goes up shortly after injury, then it goes down for few days and starts to go up again after 6-10 days after injury, during the repair period. Might be this phenomenon should be better explained and discussed? Indeed, on the same experiment they showed that activation GLP1r by its agonist reduced remarkedly reactive glia and improved brain regeneration.
AUTHOR RESPONSE: Thank you for the suggestion. We have now expanded and discussed these studies, please see the highlighted text on Pages 15-16.
REVIEWER COMMENT. More, GLP1r agonist have been shown to act as a modulators of renin angiotensin aldosterone (RAAS) pathway. Specifically, they have been shown to improve ACE2 activity, resulting the activation of compensatory axis of RAAS (Agtr2, Mas1 axis), what also improves cellular regeneration or apoptosis, reduces inflammation and fibrosis and makes blood vessels more permeable for minerals and nutrients.
AUTHOR RESPONSE: We have now added a paragraph on the significance of RAAS/Mas receptor axis in the context of neurodegeneration and GLP-1, please see the highlighted text on Pages 19-20.
REVIEWER COMMENT. On the other hand, might be the connection of GABA and GLP1 should be discussed. It seems that the activation of GLP1r increases GABA synthesis both in beta cells and in GABAeric neurons. In the pancreatic beta cells, GABA is released during insulin secretion. Insulin is increasing alpha cell GABA receptor expression and after GABA binds of it, glucagon secretion is reduced resulting the block of gluconeogenesis. On the same time GABA can bind to pro-alpha cells to induce their proliferation to beta cells. And it seems, that GABA needs GLP1r to exert its pancreatic function. Same in CNS, there have been shown that the activation of GLP1r in GABAeric neurons improves neuronal plasticity. Might be this should be also discussed.
AUTHOR RESPONSE: We have now discussed the significance of GABA in islet function and implications for GLP1 based approaches as suggested by the reviewer, please see the highlighted text on Pages 17-18.
Reviewer 2 Report
I have no comments on the written paper, and I suggest the acceptance in the present form
Author Response
REVIEWER COMMENT. I have no comments on the written paper, and I suggest the acceptance in the present form.
AUTHOR RESPONSE: We sincerely thank the reviewer the time and positive recommendation
Reviewer 3 Report
This review paper summarizes the research related to the use of GLP-1 receptor agonists for neuroprotection against neurodegenerative disorders and their future scope which is an interesting and important area of research. The paper is generally well written and merits publication; however, the quality of the paper can be enhanced if the following points can be addressed.
1. Please write the full form of abbreviations during their first usage in the manuscript as some readers might not be familiar with the abbreviated forms used. For example: Line 23 “NVU”.
2. Please include proper references for Table 1 with a separate column for references.
3. Please consider including some recent references which showed efficacy of GLP-1r agonists and DPP-4 inhibitors against neuronal inflammation in animal models. Example:
Antioxidants 2021, 10(8), 1175; https://doi.org/10.3390/antiox10081175
Antioxidants 2021, 10(9), 1438; https://doi.org/10.3390/antiox10091438
Int. J. Mol. Sci. 2022, 23(2), 739; https://doi.org/10.3390/ijms23020739
4. There is always a dilemma on how to conclude a review article. Since the authors have deliberately summarized huge amounts of published results, it will go a long way. It would be helpful if they can provide their own thoughts that would in turn help in finding the areas that need to be addressed. For example, what are the factors that one needs to consider while choosing GLP-1RA or DPP-4i for neurodegenerative disorders, and what are the steps required for the fast transition of these drugs for clinical setting.
Author Response
RESPONSES TO REVIEWER 3 COMMENTS:
We appreciate the time and effort of the reviewer in suggesting improvements to the manuscript. The following is a point-by-point response to the reviewer’s comments.
- New text in relevant sections is highlighted in green
REVIEWER 3
REVIEWER COMMENT. This review paper summarizes the research related to the use of GLP-1 receptor agonists for neuroprotection against neurodegenerative disorders and their future scope which is an interesting and important area of research. The paper is generally well written and merits publication; however, the quality of the paper can be enhanced if the following points can be addressed.
- Please write the full form of abbreviations during their first usage in the manuscript as some readers might not be familiar with the abbreviated forms used. For example: Line 23 “NVU”.
AUTHOR RESPONSE: We apologize for the omission; now have provided the full forms of the abbreviations throughout the text, as suggested by the reviewer.
- Please include proper references for Table 1 with a separate column for references.
AUTHOR RESPONSE: This information is derived from References 2 and 23, and is clearly indicated for the reader on the first citation of Table 1 (Page 4 of the main text).
- Please consider including some recent references which showed efficacy of GLP-1r agonists and DPP-4 inhibitors against neuronal inflammation in animal models. Example:
Antioxidants 2021, 10(8), 1175; https://doi.org/10.3390/antiox10081175
Antioxidants 2021, 10(9), 1438; https://doi.org/10.3390/antiox10091438
Int. J. Mol. Sci. 2022, 23(2), 739; https://doi.org/10.3390/ijms23020739
AUTHOR RESPONSE: Thank you for the suggestions, we have now added these studies/opinions as References 205, 285 and 241 respectively in the revised text. Also, we have discussed their salient points (highlighted text) on Page 14 and Page 20 in relevant sections.
4 . There is always a dilemma on how to conclude a review article. Since the authors have deliberately summarized huge amounts of published results, it will go a long way. It would be helpful if they can provide their own thoughts that would in turn help in finding the areas that need to be addressed. For example, what are the factors that one needs to consider while choosing GLP-1RA or DPP-4i for neurodegenerative disorders, and what are the steps required for the fast transition of these drugs for clinical setting.
AUTHOR RESPONSE: We appreciate the suggestion on expanding the conclusion section with our own viewpoints on the clinical transition. Please see the highlighted text on Pages 21-22.